# Shedding of infectious SARS-CoV-2 despite vaccination

**Kasen K. Riemersma[1], Luis A. Haddock, III[1], Nancy A. Wilson[2], Nicholas Minor[2], Jens Eickhoff[3], Brittany E. Grogan[4], Amanda Kita-Yarbro[4], Peter J. Halfmann[1], Hannah E. Segaloff[5], Anna Kocharian[6], Kelsey R. Florek[7], Ryan Westergaard[8], Allen Bateman[7], Gunnar E. Jeppson[9], Yoshihiro Kawaoka[1], David H. O'Connor[2☯, Thomas C. Friedrich[1]☯, Katarina M. Grande[4]☯ ***

1 Department of Pathobiological Sciences, University of Wisconsin School of Veterinary Medicine, Madison, Wisconsin, United States of America, 2 Department of Pathology and Laboratory Medicine, University of Wisconsin School of Medicine and Public Health, Madison, Wisconsin, United States of America, 3 Biostatistics and Medical Informatics, University of Wisconsin School of Medicine and Public Health, Madison, Wisconsin, United States of America, 4 Public Health Madison & Dane County, Madison, Wisconsin, United States of America, 5 Epidemic Intelligence Service, CDC, Atlanta, Georgia, United States of America, 6 Wisconsin Department of Health Services, Madison, Wisconsin, United States of America, 7 Wisconsin State Laboratory of Hygiene, Madison, Wisconsin, United States of America, 8 Department of Medicine, University of Wisconsin School of Medicine and Public Health, Madison, Wisconsin, United States of America, 9 Exact Sciences, Madison, Wisconsin, United States of America

☯ These authors contributed equally to this work.
* KGrande@publichealthmdc.com

**Data Availability Statement:** Data and processing workflows are available at https://go.wisc.edu/p22l16. To protect potentially personally identifiable information, the publicly available dataset contains only PCR Ct values, vaccine

## Abstract

The SARS-CoV-2 Delta Variant of Concern is highly transmissible and contains mutations that confer partial immune escape. The emergence of Delta in North America caused the first surge in COVID-19 cases after SARS-CoV-2 vaccines became widely available. To determine whether individuals infected despite vaccination might be capable of transmitting SARS-CoV-2, we compared RT-PCR cycle threshold (Ct) data from 20,431 test-positive anterior nasal swab specimens from fully vaccinated (n = 9,347) or unvaccinated (n = 11,084) individuals tested at a single commercial laboratory during the interval 28 June– 1 December 2021 when Delta variants were predominant. We observed no significant effect of vaccine status alone on Ct value, nor when controlling for vaccine product or sex. Testing a subset of low-Ct (<25) samples, we detected infectious virus at similar rates, and at similar titers, in specimens from vaccinated and unvaccinated individuals. These data indicate that vaccinated individuals infected with Delta variants are capable of shedding infectious SARS-CoV-2 and could play a role in spreading COVID-19.

## Author summary

A pivotal moment in the COVID-19 pandemic in the U.S. occurred during the summer of 2021—after the majority of people were vaccinated against the virus that causes COVID. The paradigm at the time was that infection and transmission after vaccination were rare. After contact tracers noticed an increase in infections after vaccination, we rapidly

status, age, sex,manufacturer, symptom status, virus culture status, and days from symptom onset to testing for each specimen.

**Funding:** This work was supported by Centers for Disease Control and Prevention contracts 75D30120C09870 and 75D30121C11060 to D.H.O and T.C.F. The authors are also members of the Upper Midwest Regional Accelerator for Genomic Surveillance funded by the Rockefeller Foundation. The funders had no role in study design, data collection and analysis, decision to publish, or preparation of the manuscript. The authors did not receive salary from the funders.

**Competing interests:** I have read the journal's policy and the authors of this manuscript have the following competing interests: Amanda Kita-Yarbro owns 25 shares of Pfizer stock. The other authors have declared that no competing interests exist.

assembled a team of virologists, epidemiologists, and public health officials to investigate. Our study was conducted in Wisconsin at a time when the Delta variant accounted for almost all new infections. While data related to individual outbreaks and large gatherings were emerging, we examined data from community test sites spread over a wide geographic area in Wisconsin. We found that a large proportion of people with infection despite full vaccination had high levels of virus in their bodies, regardless of sex or the type of vaccine they received. Our study was one of the first to demonstrate the possibility that vaccinated people could play a role in spreading COVID, and helped inform public health policies (such as mask mandates) to cope with new surges in COVID-19 cases.

## Introduction

The SARS-CoV-2 Delta variant was initially characterized in March 2021 and was associated with increased infection incidence in North America beginning in the summer of 2021. In Wisconsin, Delta-lineage viruses were first detected on 12 April 2021, and within 10 weeks accounted for more than 90% of sequenced viruses. Delta viruses were highly transmissible and contained mutations that confer partial immune escape. The "surge" in cases attributable to Delta-lineage viruses represented the first substantial increase in SARS-CoV-2 infection incidence after vaccines had become widely available in the United States. By July 2021, SARS-CoV-2 infection incidence was low in the United States (*https://www.cdc.gov/coronavirus/2019-ncov/covid-data/covidview/past-reports/05212021.html#print*) [1], and national and local public health agencies were loosening requirements for face coverings and other non-pharmaceutical interventions to reduce virus transmission [1–3]. A key question in developing these policies was whether persons infected with SARS-CoV-2 despite vaccination could transmit infection to others.

By late July 2021, outbreak investigations suggested that vaccinated persons who became infected could spread Delta-lineage SARS-CoV-2 [4,5]. To determine whether individuals with vaccine breakthrough infections could shed Delta viruses at levels consistent with potential transmission, we compared the SARS-CoV-2 RNA burden in nasal swab specimens from vaccinated and unvaccinated individuals tested at a single commercial laboratory. We also attempted virus isolation and determined infectious viral titers from a subset of samples from vaccinated and unvaccinated individuals.

## Methods

### Ethics statement

The University of Wisconsin-Madison Institutional Review Board deemed that this project qualifies as public health surveillance activities as defined in the Common Rule, 45 CFR 46.102 (l)(2). As such, the project is not deemed to be research regulated under the Common Rule and therefore, does not require University of Wisconsin-Madison IRB review and oversight.

### Study design

This was a cross-sectional study analyzing residual test-positive anterior nasal swab specimens submitted for clinical testing to a single commercial RT-PCR testing provider between 28 June 2021 and 1 December 2021. The study objective was to use sample-associated metadata to determine whether vaccination status impacted the estimated viral RNA burden in individuals testing positive for SARS-CoV-2 during the time of high Delta variant prevalence. The cutoff

date was chosen to exclude samples containing the Omicron variant, which was first detected in Wisconsin on 4 December 2021. The estimated prevalence of Delta in Wisconsin was 60% at the start of the study on 28 June 2021, reached 95% by 23 July 2021, and remained at or above 95% until 12 December, 2021 (outbreak.info). Samples were collected using standardized collection kits from individuals seeking SARS-CoV-2 RT-PCR testing at multiple clinic locations across the state of Wisconsin. To estimate nasal viral RNA burden, we compared RT-PCR cycle threshold (Ct) data from 20,431 specimens from fully vaccinated (n = 9,347) or unvaccinated (n = 11,084) individuals. All viral RNA extraction and RT-PCR was performed at Exact Sciences in Madison, WI using the same protocol.

## RT-PCR assay

The Flu-SC2 Multiplex Assay (https://www.cdc.gov/coronavirus/2019-ncov/lab/multiplex.html)) as implemented by Exact Sciences was used to determine Ct values. This RT-PCR assay can simultaneously detect nucleic acid from SARS-CoV-2, as well as Influenza A and B from anterior nasal swabs. RNA extraction was conducted using Exact Sciences Corporation's proprietary extraction procedure on the Hamilton STARlet liquid handler. The oligonucleotide primers and probe for detection of SARS-CoV-2 were selected from an evolutionarily conserved region of the 3' terminus of SARS-CoV-2 genome and also cover part of the 3'-terminal portion of the nucleocapsid (N) gene. RNA isolated from anterior nasal swab specimens was reverse transcribed into cDNA and amplified using the ThermoFisher TaqPath 1-Step RT-qPCR Master Mix and Applied Biosystems 7500 Fast Dx Real-Time PCR Instrument with SDS version 1.4.1 software. Controls included a no-template control, a positive extraction control containing human RNAse P, and an internal control for RNAse P.

## Defining vaccination status

Individuals were considered fully vaccinated at the time of testing if vaccine registry or self-reported data indicated receiving a final vaccine dose at least 14 days prior to submitting the specimen that tested positive for SARS-CoV-2 and was used in our analysis. We used validated public health vaccine registries for the State of Wisconsin where possible. Self-reported vaccination status was included with sample metadata submitted by testing providers to the Exact Sciences laboratory; when individuals' vaccination status was not available in public health databases, we used self-report data to determine status. Comparing self-reporting to data from vaccine registries determined that under-reporting of full vaccination status was more common than over-reporting (S1 Fig).

Specimens from individuals who were partially vaccinated (i.e., had not received a complete vaccine series, were tested <14 days after the final dose, or those whose vaccination dates were after the sample collection date) were excluded. We also excluded 430 samples from individuals who received a booster vaccine dose prior to the sample collection date, since these individuals represented a small fraction of the total number of available samples and booster effects could confound our analyses.

## Virus isolation and plaque assay

With an initial set of specimens with Ct values <25, we assessed the presence of infectious virus by inoculating residual specimens onto a monolayer of Vero E6/TMPRSS2 cells and monitoring for the presence of cytopathic effects over 5 days. Specimens were selected by N1 Ct-matching between fully vaccinated and unvaccinated persons. Specimens from individuals with unknown vaccine status were excluded from this assay. With a second set of samples, we determined virus titer, expressed as plaque-forming units (PFU) per ml specimen, by using a

10-fold dilution series along with undiluted samples to infect a monolayer of Vero E6/TMPRSS2 cells (100 µl per well) for 30 minutes at 37˚C. The cells were washed once to remove unbound virus, then overlaid with 1% methylcellulose for four days at which time plaques were counted.

## Statistical analysis

We used analysis of variance (ANOVA) to evaluate how Ct values varied between age groups, sexes, and by vaccine product, as well as two-way interactions between these factors. Raw Ct values were not normally distributed, so we log-transformed all Ct values prior to ANOVA, and confirmed normality by plotting residuals and normal probability (S2 Fig). We report least square means along with the corresponding 95% confidence intervals (CIs). Tukey's Honestly Significant Difference Method (HSD) was used to control the type I error when conducting multiple comparisons between groups. Because our dataset included individuals with varying amounts of time between vaccination and SARS-CoV-2 infection, it is possible that waning levels of immunity could impact susceptibility to infection and/or viral loads after vaccination. To determine whether there was a relationship between time since vaccination and Ct values in infected persons, we conducted additional regression analyses that included months since completion of vaccination as a vaccine manufacturer-specific continuous predictor variable. Months since completion of vaccination was defined as the number of days since completion divided by 30.44, the average number of days per month.

In order to quantify and interpret differences between groups, we calculated standardized differences (Cohen's effect size $d$), defined as the mean differences between groups divided by the pooled standard deviations. Effect sizes of $d<0.2$ were considered to indicate either no difference or a negligible difference between populations. An effect size of 0.2 to 0.5 indicated a small difference, 0.5 to 0.8 was a moderate difference, and $>0.8$ was a large difference. The proportions of subjects with Ct values $<25$ were compared between groups using a chi-square test.

The results of the primary comparisons were confirmed by conducting nonparametric analyses. Specifically, the nonparametric Wilcoxon rank sum test was used to conduct comparisons between Ct values between the two groups, and the nonparametric Kruskal-Wallis test was used to conduct the comparisons of Ct values between more than two groups. Statistical analyses were conducted using SAS software (SAS Institute, Cary NC), version 9.4, figures were plotted using the R package ggplot2 [6] or from Prism version 9.3.1.

## Results

### Individuals infected with SARS-CoV-2 despite vaccination have low Ct values

SARS-CoV-2 RT-PCR Ct values $<25$ had previously been associated with shedding of infectious SARS-CoV-2 [7,8]. We observed Ct values $<25$ in 6,253 of 9,347 fully vaccinated (67%) and 6,739 of 11,084 (61%) unvaccinated individuals (Fig 1A). Because of the very large number of samples, very small differences in outcome variables may nonetheless reach statistical significance when using p values with a traditional alpha set to 0.05. That is, we may find small differences between groups that are statistically significant ($p < 0.05$), but have a negligible effect ($d < 0.2$). In order to quantify the magnitude of differences between groups, we calculated standardized differences (Cohen's effect size $d$), defined as the mean differences between the groups divided by the pooled standard deviations. A value of $d < 0.2$ indicates negligible effects of the analyzed variables on the outcome variable. Here we report values for both p and

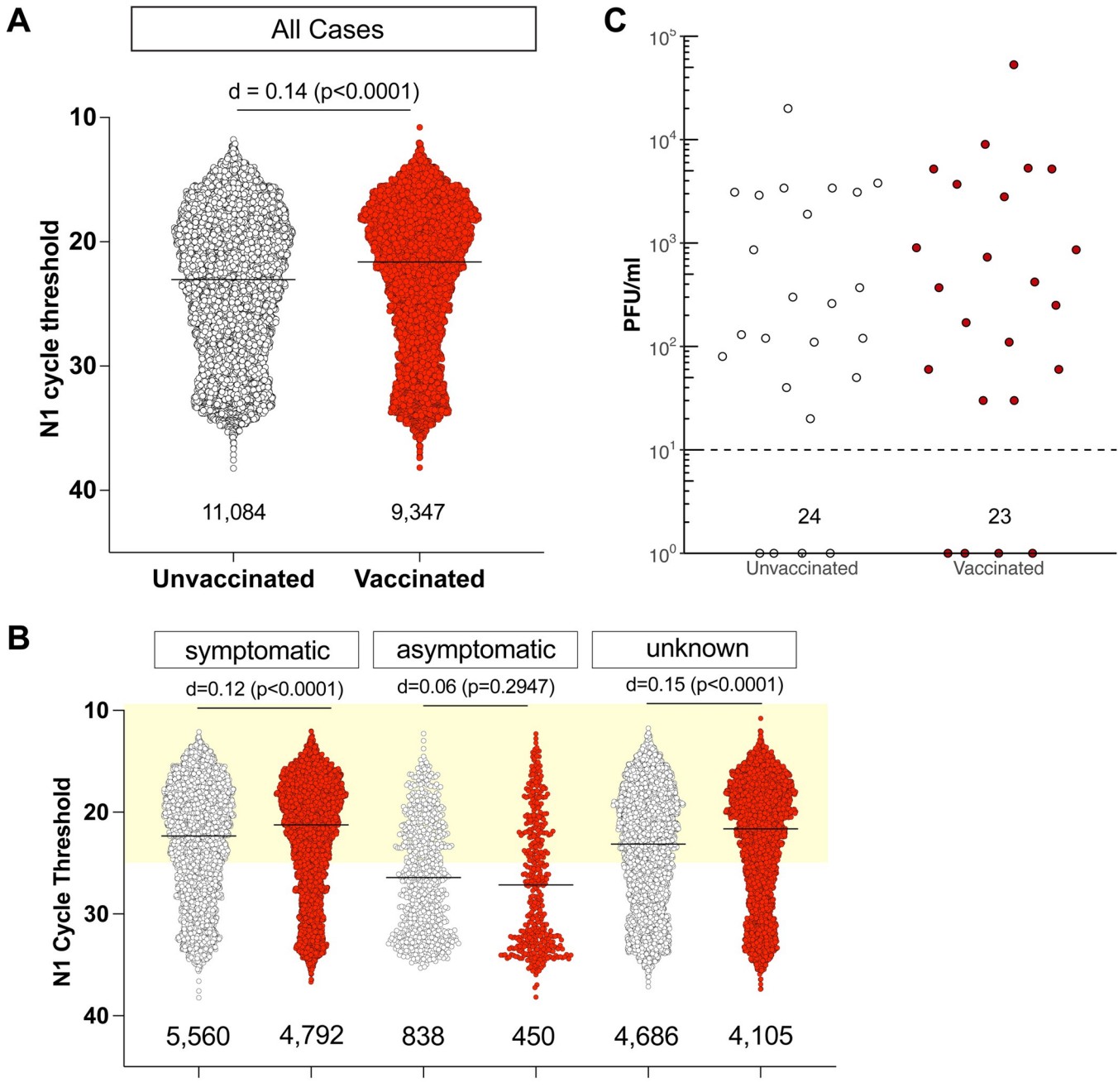

**Fig 1. Individuals infected with SARS-CoV-2 despite full vaccination have low Ct values and shed similar amounts of infectious virus as unvaccinated individuals. A.** N1 Ct values for SARS-CoV-2-positive specimens were grouped by vaccination status. RT-PCR was performed by Exact Sciences Corporation, responsible for over 10% of all PCR tests in Wisconsin during this period, using a qualitative diagnostic assay targeting the SARS-CoV-2 N gene (oligonucleotides identical to CDC's N1 primer and probe set) that has been authorized for emergency use by FDA (*https://www.fda.gov/media/138328/ download*)). See also **Table 1**. An effect size of *d*< 0.2 is negligible. The number of samples in each group is listed under the dot plot. **B.** N1 Ct values for SARS-CoV-2-positive specimens grouped by vaccination status for individuals who were symptomatic or either asymptomatic or did not have any information, at the time of testing. Light yellow box indicates Ct values <25. **C.** We performed plaque assays on Vero E6 TMPRSS2 cells on a subset of specimens. Specimens were selected by N1 Ct-matching between fully vaccinated and unvaccinated persons. Specimens from individuals with unknown vaccination status were excluded from the analysis. Infectious titers are expressed as plaque-forming units (PFU) per milliliter of specimen. Specimens underwent a freeze-thaw cycle prior to virus titration.

*d* for completeness. We observed no significant effect of vaccination status on Ct values in infected persons (Cohen's *d* = 0.14, p<0.0001; **Table 1**). Low Ct values were detected in vaccinated people whether or not they reported symptoms at the time of testing (**Fig 1B**), with Ct values <25 detected in 65% (95% CI:63–66%) of unvaccinated symptomatic individuals and in 70% (95% CI:69–71%) of fully vaccinated symptomatic individuals (p<0.0001). Notably, for symptomatic individuals, time from symptom onset to testing did not vary by vaccination status. Both vaccinated and unvaccinated individuals in our population reported a median time of 2.4 days between symptom onset and testing. 92% of individuals in our dataset sought testing within 6 days of symptom onset. Together these results suggest that our observations are not confounded by biases in test-seeking behavior between vaccinated and unvaccinated persons (Two-sided K-S test: p = 0.0012; medians 2.40d unvaccinated, 2.42d vaccinated, **S3 Fig**).

## Individuals infected with SARS-CoV-2 despite vaccination shed infectious virus

Previous studies focusing primarily on unvaccinated individuals suggested that RT-qPCR Ct values <25 may be strongly associated with the shedding of infectious SARS-CoV-2 [8,9]. To determine whether vaccinated persons with potentially high viral burdens might be capable of shedding infectious virus, we inoculated a subset of residual specimens with Ct values <25 onto a monolayer of Vero E6/TMPRSS2 cells and monitored for the presence of cytopathic effects over 5 days. Specimens were selected by N1 Ct-matching between fully vaccinated and unvaccinated persons. Specimens from individuals with unknown vaccine status were excluded from this assay. 37 of 39 specimens from vaccinated individuals contained culturable SARS-CoV-2, as compared with 15 of 17 specimens from unvaccinated persons (**S4 Fig**). We therefore performed virus titration on a second set of samples with Ct < 25 and found no difference in infectious virus titer between samples from vaccinated vs. unvaccinated individuals (**Fig 1C**).

## Ct value in breakthrough infection is not strongly affected by vaccine product, age, or sex

We considered whether different vaccine products affected Ct values observed in individuals with breakthrough infections. Vaccination had neglible effects on mean Cts in vaccinated as compared with unvaccinated individuals, regardless of the manufacturer, (Janssen (JNJ-78436735) effect size *d* = 0.18, p<0.0001; Moderna (mRNA-1273) effect size *d* = 0.07, p = 0.0052; Pfizer (BNT162b2) effect size *d* = 0.17, p<0.0001; **S5A Fig**; see also **S1 Table**). Low-Ct samples were found in similar proportions among all groups, Janssen 68% Ct<25, Moderna 64% Ct<25 and Pfizer 68% Ct<25.

   Vaccine effectiveness, particularly against symptomatic, test-positive SARS-CoV-2 infection, wanes with time after vaccine receipt [10–21]. We therefore asked whether Ct values decreased as a function of time between last vaccination and the time at which individuals tested positive for SARS-CoV-2 infection. Indeed, when considering all vaccine products combined, there was

**Table 1. Vaccinated vs. Unvaccinated.**

|  | Means | 95% CI | Effect size *d* | p-value |
|---|---|---|---|---|
| Unvaccinated (N = 11,084) | 22.9 | 22.8–23.0 | 0.14 | <0.0001 |
| Vaccinated (N = 9,347) | 22.1 | 22.0–23.2 |  |  |

Interpretation of effect size d: (d<0.2 no difference/negligible difference, 0.2–0.5 small difference, 0.5–0.8 moderate difference, >0.8 large difference)

a small, but statistically significant decrease in Ct values (consistent with higher levels of SARS-CoV-2 RNA in swab specimens) as the time between last vaccination and positive test increased (Slope: -0.18, 95% CI: -0.26–0.10; p-value<0.0001; **S5B Fig**). However, when we stratify individuals according to vaccine product received, we find that this effect seems to be driven principally by high Ct values among Pfizer vaccine recipients infected in the first month after vaccination, as the slopes of Ct value vs time between vaccination and infection are not significantly different from zero for recipients of the other two products (**S5B Fig**).

Age and male sex have been considered risk factors for COVID-19 disease [22–26]. While one might hypothesize that older individuals and/or males might have higher SARS-CoV-2 burdens and therefore lower Ct values at the time of testing, evidence for this is mixed, with some studies reporting lower Ct values in older individuals [24,27], others in younger individuals [28], and still others finding no difference by age [20,29–34]. We therefore stratified groups based on age and compared Ct values by age group. Vaccination status had negligible effects on Ct values ($d<0.2$) for all age groups considered except those aged 0–11 years (**S2 Table**). In this group, there were very few vaccinated individuals (N = 7), as would be expected because vaccines had not been approved for those 11 and under for most of our study period. Therefore, despite the significant effect size ($d = 0.79$, p = 0.0466), we do not believe our data strongly support the notion that vaccination status has a strong effect on Ct value in children under 12. When comparing Ct values between unvaccinated and vaccinated within males and females, negligible differences were observed (female: $d = 0.14$, male: $d = 0.15$; **S3 Table**).

## Discussion

The emergence of Delta variants in the United States led to the first wave of increasing case burdens following the widespread availability of SARS-CoV-2 vaccines. At the time, prevailing public health recommendations were that vaccinated persons need not use face coverings in indoor settings. These recommendations were based in part on the fact that vaccines demonstrated remarkable effectiveness against test-positive SARS-CoV-2 infection in initial clinical trials conducted in 2020 [35–40], suggesting that vaccinated persons might play negligible roles in SARS-CoV-2 transmission. However, the initial vaccine effectiveness studies were conducted when ancestral variants predominated, prior to the emergence of variants of concern. Here we conducted a comprehensive retrospective analysis of RT-PCR Ct values in persons infected with SARS-CoV-2 during the time when Delta variants predominated, to determine whether individuals infected with Delta variants despite vaccination could be involved in community spread of SARS-CoV-2. Combined with other studies [41,42] our data indicate that vaccinated as well as unvaccinated individuals infected with SARS-CoV-2 Delta variants can shed, and potentially transmit, infectious virus [43,44]. We find low Ct values in substantial proportions of both unvaccinated and vaccinated individuals who tested positive for SARS-CoV-2 during the time when Delta variants predominated, in agreement with other recent reports [41,44–47]. The occurrence in our dataset of positive samples from multiple Wisconsin counties without a linking outbreak (more than 80% of samples were not associated with an outbreak known to public health) indicate that Delta-lineage SARS-CoV-2 can achieve low Ct values consistent with transmissibility in fully vaccinated individuals across a range of environments. Importantly, we also show that infectious SARS-CoV-2 is found at similar titers in vaccinated and unvaccinated persons. An important limitation of our study is that we analyzed only single specimens from each infected individual, so our data cannot determine whether vaccinated individuals control virus replication in the upper respiratory tract more quickly than unvaccinated persons, as other studies have suggested [42]. We also note that the duration and level of infectious virus shedding varies widely among individuals [48], and that Ct

values are an imperfect proxy for shedding of infectious virus. However, the vast majority of individuals included in our study were tested within 6 days of symptom onset (S3 Fig), a time before viral loads diverged in vaccinated and unvaccinated persons tested daily in a previous study [42]. Our cross-sectional, laboratory-based study was also not designed to detect or quantify differences in the relative roles of vaccinated and unvaccinated persons in spreading SARS-CoV-2 in the community. We find that a substantial proportion of individuals infected with Delta viruses despite vaccination had low Ct values consistent with the potential to shed infectious virus. Our findings support the notion that persons infected despite vaccination can transmit SARS-CoV-2. Therefore, preventing infection is critical to preventing transmission. Vaccinated and unvaccinated persons should be tested when symptomatic or after close contact with someone with suspected or confirmed COVID-19. Continued adherence to non-pharmaceutical interventions during periods of high community transmission to mitigate spread of COVID-19 remains important for both vaccinated and unvaccinated individuals.

## Supporting information

**S1 Fig. Concordance between self-reported vaccination status and records in public health vaccine registries.** Individuals were considered fully vaccinated based on vaccine registry (WIR/WEDSS) data if the registries indicated receipt of a final vaccine dose at least 14 days prior to submitting the sample used in our analysis. For individuals whose vaccination status could not be verified in the registry, self-reported data collected at the time of testing were used. Individuals were considered unvaccinated based on self-report only if there was an explicit declaration of unvaccinated status in the self-reported data. Individuals were considered fully vaccinated based on self-report if they fulfilled all of the following criteria: (1) indicated that they had received a COVID vaccine prior to testing; (2) indicated that they did not require another vaccine dose; and (3) reported a date of last vaccine dose that was at least 14 days prior to testing. Specimens lacking data on vaccination status were excluded from the study. Specimens from partially vaccinated individuals (incomplete vaccine series, or <14 days post-final dose) were also excluded. Specimens from individuals who received a booster prior to sample collection were also excluded as non-equivalent to those fulfilling the criteria to be considered fully vaccinated. **A.** Of 20,431 specimens with vaccination status available from at least one source, 5,078 specimens had data available from both sources. Under-reporting of full vaccination status in self-reports 1,064/6,142 or 17%) was more common than over-reporting (409/5,487 or 7.4%). **B.** N1 Ct values for SARS-CoV-2-positive specimens grouped by vaccination status for individuals whose vaccination status was determined by vaccine registry or by self-reported data.
(TIF)

**S2 Fig. Log transformation of raw Ct values results in normally distributed residuals.** Raw Ct values were not normally distributed, so we log-transformed all Ct values prior to ANOVA, and confirmed normality by plotting residuals and normal probability.
(TIF)

**S3 Fig. Density distributions of unvaccinated and vaccinated specimen collection dates by day since symptom onset.** Day 0 on the x-axis denotes self-reported day of symptom onset. Negative values for days indicate specimen collection prior to symptom onset. Symptom onset data were available for n = 6,871 unvaccinated cases and n = 5,522 vaccinated cases. Two-sided K-S test: p = 0.0012; median days since symptom onset were 2.4 for both unvaccinated and vaccinated cases.
(TIF)

**S4 Fig. Infectious SARS-CoV-2 detected in the majority of fully vaccinated individuals with low Ct values.** Infectiousness was determined for a subset of N1 Ct-matched specimens with Ct <25 by inoculation onto Vero E6 TMPRSS2 cells, then determining presence or absence of cytopathic effects (CPE) after 5 days in culture. Specimens with unknown vaccination status were excluded from the analysis. Circles indicate presence of CPE; 'X' indicates no CPE detected.
(TIF)

**S5 Fig. Ct values do not differ substantially by vaccine type. A.** Comparison of mean N1 Ct values in all specimens, stratified by vaccine type shows negligible effect ($d < 0.2$) of vaccine type on Ct value at time of positive test, relative to unvaccinated persons. **B.** The time analysis showed a decrease in N1 Ct values with time over 7 months. Combining all three vaccines, there was a significant decrease over the first 7 months, with a slope of -0.18 (95% CI: -0.26–0.10), p value <0.0001. Individually, Janssen had a slope -0.19 (95% CI: -0.38 to -0.001, p-value = 0.060), Moderna had a slope of -0.13 (95% CI: -0.28–0.02, p-value = 0.092), Pfizer had a slope of -0.24 (95% CI: -0.24 to -0.13, p-value<0.0001).
(TIF)

**S1 Table. Comparisons of Ct values between vaccine types.** Vaccination had negligible effects on mean Cts in vaccinated as compared with unvaccinated individuals, regardless of the vaccine manufacturer.
(DOCX)

**S2 Table. Comparison of Ct values in vaccinated and unvaccinated persons, stratified by age group.** Vaccination status had negligible effects on Ct values ($d<0.2$) for all age groups considered except those aged 0–11 years—there is a significant interaction between age group and vaccination status, p<0.0001. However, in this group, there were very few vaccinated individuals (N = 7), as would be expected because vaccines had not been approved for those 11 and under for most of our study period. Therefore, despite the significant effect size, we do not believe our data strongly support the notion that vaccination status has a strong effect on Ct value in children under 12.
(DOCX)

**S3 Table. Comparison of Ct values in vaccinated and unvaccinated persons, stratified by sex.** When comparing Ct values between unvaccinated and vaccinated within males and females, negligible differences were observed.
(DOCX)

## Acknowledgments

The opinions expressed by authors contributing to this journal do not necessarily reflect the opinions of the Centers for Disease Control and Prevention or the institutions with which the authors are affiliated.

## Author Contributions

**Conceptualization:** Kasen K. Riemersma, Luis A. Haddock, III, Nancy A. Wilson, Nicholas Minor, Ryan Westergaard, David H. O'Connor, Thomas C. Friedrich, Katarina M. Grande.

**Data curation:** Nancy A. Wilson, Nicholas Minor, Brittany E. Grogan, Amanda Kita-Yarbro, Gunnar E. Jeppson.

**Formal analysis:** Kasen K. Riemersma, Nancy A. Wilson, Nicholas Minor, Jens Eickhoff, Hannah E. Segaloff, Anna Kocharian.

**Funding acquisition:** David H. O'Connor, Thomas C. Friedrich.

**Investigation:** Kasen K. Riemersma, Peter J. Halfmann, Yoshihiro Kawaoka, David H. O'Connor, Thomas C. Friedrich.

**Methodology:** Kasen K. Riemersma, Luis A. Haddock, III, Nancy A. Wilson, Nicholas Minor, Ryan Westergaard, Yoshihiro Kawaoka, David H. O'Connor, Thomas C. Friedrich.

**Project administration:** David H. O'Connor, Thomas C. Friedrich.

**Resources:** Yoshihiro Kawaoka, David H. O'Connor, Thomas C. Friedrich.

**Software:** Luis A. Haddock, III, Nancy A. Wilson, Nicholas Minor.

**Supervision:** Nancy A. Wilson, David H. O'Connor, Thomas C. Friedrich, Katarina M. Grande.

**Validation:** Hannah E. Segaloff, Anna Kocharian.

**Visualization:** Kasen K. Riemersma, Nancy A. Wilson, Jens Eickhoff.

**Writing – original draft:** Kasen K. Riemersma, David H. O'Connor, Thomas C. Friedrich, Katarina M. Grande.

**Writing – review & editing:** Kasen K. Riemersma, Nancy A. Wilson, Nicholas Minor, Jens Eickhoff, Brittany E. Grogan, Amanda Kita-Yarbro, Peter J. Halfmann, Hannah E. Segaloff, Anna Kocharian, Kelsey R. Florek, Ryan Westergaard, Allen Bateman, Gunnar E. Jeppson, Yoshihiro Kawaoka, David H. O'Connor, Thomas C. Friedrich, Katarina M. Grande.

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
