## [Decision Letter · Decision Letter 0]

17 Feb 2022

Dear Ms. Grande,

Thank you very much for submitting your manuscript "Shedding of Infectious SARS-CoV-2 Despite Vaccination" for consideration at PLOS Pathogens. As with all papers reviewed by the journal, your manuscript was reviewed by members of the editorial board and by several independent reviewers. In light of the reviews (below this email), we would like to invite the resubmission of a significantly-revised version that takes into account the reviewers' comments.

We cannot make any decision about publication until we have seen the revised manuscript and your response to the reviewers' comments. Your revised manuscript is also likely to be sent to reviewers for further evaluation.

Sincerely,

Florian Krammer, PhD

Associate Editor

PLOS Pathogens

Kanta Subbarao

Section Editor

PLOS Pathogens

Kasturi Haldar

Editor-in-Chief

PLOS Pathogens

orcid.org/0000-0001-5065-158X

Michael Malim

Editor-in-Chief

PLOS Pathogens

orcid.org/0000-0002-7699-2064

Reviewer's Responses to Questions

**Part I - Summary**

Reviewer #1: The authors present an interesting study in which they investigate the ability of unvaccinated and vaccinated individuals infected with SARS-CoV-2 to shed virus by quantifying virus in nasal swabs. While this work is important, and the effectiveness of vaccination against SARS-CoV-2 variants of concern is topical, the manuscript suffers from several shortfalls.

The authors report that the swabs obtained from vaccinated which are defined as “having received a second mRNA vaccine dose or single adenovirus vector vaccine dose ≥ 2 weeks prior to testing positive”. The authors do not differentiate between patients which have received different vaccines, nor do they differentiate swabs based on the precise time after vaccination in which these individuals have tested positive. Given the published differences in protection which are elicited from the different vaccines (RNA vaccines vs Adv5 vaccines for example), it is difficult to compare between these groups and form conclusions on the ability of delta to infect vaccinated individuals. Additionally, it has been shown that peak titres are achieved around 4 weeks following booster vaccination and that these steadily decline over the following months. As the samples derived from vaccinated individuals may have been sampled anywhere from 2 weeks after vaccination to several months after vaccination, no strong conclusion can be made on the effectiveness of the vaccines to protect against viral shedding. Due to these confounding factors, it is not possible to make any robust conclusions from the data provided. Unless the authors can retrieve meta data regarding the vaccine received by the subjects and more precise data on the timing of the infection following vaccination, this study is not publishable.

Furthermore, the methods provided by the authors lacks detail. No information is given regarding the RNA extraction from the swabs, the PCR methodology used, the primers used, the PCR conditions employed, how plaque assays were carried out, or details on the statistical analysis employed.

Reviewer #2: (No Response)

**Part II – Major Issues: Key Experiments Required for Acceptance**

Reviewer #1: In order to render this manuscript publishable, the data must be sorted according to the type of vaccine recieved by the vaccinated indviduals in addition to the time since vaccination. This would allow for comparisons between the vaccine types and provide data on the effectiveness of the vaccines over time, Further confounding factors such as age and sex should also be investigated to determine if this biases the data.

Reviewer #2: This well-written article by Dr. Riemersma et al. examines viral shedding of the delta SARS-CoV-2 variant in vaccinated and unvaccinated individuals. The article is clear, concise, and draws appropriate conclusions from it’s findings.

**Part III – Minor Issues: Editorial and Data Presentation Modifications**

Reviewer #1: (No Response)

Reviewer #2: I only have two very minor comments:

1) Methods should have one sentence on how p-values were calculated.

2) P-values should be included in the text and abstract as appropriate.

PLOS authors have the option to publish the peer review history of their article (what does this mean?). If published, this will include your full peer review and any attached files.

Reviewer #1: No

Reviewer #2: No
---

## [Decision Letter · Decision Letter 1]

6 Aug 2022

Dear Ms. Grande,

Thank you very much for submitting your manuscript "Shedding of Infectious SARS-CoV-2 Despite Vaccination" for consideration at PLOS Pathogens. As with all papers reviewed by the journal, your manuscript was reviewed by members of the editorial board and by several independent reviewers. The reviewers appreciated the attention to an important topic. Based on the reviews, we are likely to accept this manuscript for publication, providing that you modify the manuscript according to the reviewer's specific recommendations.

Sincerely,

Florian Krammer, PhD

Associate Editor

PLOS Pathogens

Kanta Subbarao

Section Editor

PLOS Pathogens

Kasturi Haldar

Editor-in-Chief

PLOS Pathogens

orcid.org/0000-0001-5065-158X

Michael Malim

Editor-in-Chief

PLOS Pathogens

orcid.org/0000-0002-7699-2064

Reviewer Comments (if any, and for reference):

Reviewer's Responses to Questions

**Part I - Summary**

Reviewer #1: The authors have went to considerable lengths to address my previous comments. I believe the results they present are rigorous and convincing, and that the included data elevates the manuscript. All of my points have been more than adequately addressed.

Reviewer #2: Overall the authors have done a very nice job responding to the reviewers concerns. In particular, it does not appear that age or sex or other examined variables introduced bias of concern into the study. However, the methods, in particular the study design area could still be clarified. The authors should consult STROBE guidelines for a cross-sectional study.

**Part II – Major Issues: Key Experiments Required for Acceptance**

Reviewer #1: No major issues.

Reviewer #2: (No Response)

**Part III – Minor Issues: Editorial and Data Presentation Modifications**

Reviewer #1: No minor issues.

Reviewer #2: To clarify the study design methods the authors should:

1) State clearly at the beginning of the study design section that this is a cross-sectional study.

2) Provide some detail to clarify the setting. That is, where the samples were collected and why. One to two sentences should suffice.

3) I recommend that the dates sentence be moved from the last sentence of the intro into the methods.

4) Not use the word “cohort” in the study design section when they refer to the participants in their study.

PLOS authors have the option to publish the peer review history of their article (what does this mean?). If published, this will include your full peer review and any attached files.

Reviewer #1: No

Reviewer #2: No

Figure Files:

Data Requirements:

Reproducibility:

References:

---

## [Editor Report · Decision Letter 2]

12 Sep 2022

Dear Ms. Grande,

We are pleased to inform you that your manuscript 'Shedding of Infectious SARS-CoV-2 Despite Vaccination' has been provisionally accepted for publication in PLOS Pathogens.

Best regards,

Florian Krammer, PhD

Associate Editor

PLOS Pathogens

Kanta Subbarao

Section Editor

PLOS Pathogens

Kasturi Haldar

Editor-in-Chief

PLOS Pathogens

orcid.org/0000-0001-5065-158X

Michael Malim

Editor-in-Chief

PLOS Pathogens

orcid.org/0000-0002-7699-2064
---

## [Editor Report · Acceptance letter]

27 Sep 2022

Dear Ms. Grande,

We are delighted to inform you that your manuscript, "Shedding of Infectious SARS-CoV-2 Despite Vaccination," has been formally accepted for publication in PLOS Pathogens.

Best regards,

Kasturi Haldar

Editor-in-Chief

PLOS Pathogens

orcid.org/0000-0001-5065-158X

Michael Malim

Editor-in-Chief

PLOS Pathogens

orcid.org/0000-0002-7699-2064